# Identification, Characterization, and Control of Black Spot on Chinese Kale Caused by *Sphaerobolus cuprophilus* sp. nov.

**DOI:** 10.3390/plants12030480

**Published:** 2023-01-19

**Authors:** Pancheewa Kalayanamitra, Kal Kalayanamitra, Sutasinee Nontajak, Paul W. J. Taylor, Nuchnart Jonglaekha, Boonsom Bussaban

**Affiliations:** 1Department of Biology, Faculty of Science, Chiang Mai University, Chiang Mai 50200, Thailand; 2Program of Postharvest Technology, Faculty of Engineering and Agro-Industry, Maejo University, Chiang Mai 50290, Thailand; 3Royal Project Agricultural Research and Development Center, Chiang Mai 50100, Thailand; 4Faculty of Science, The University of Melbourne, Parkville, VIC 3010, Australia; 5Research Center of Microbial Diversity and Sustainable Utilization, Faculty of Science, Chiang Mai University, Chiang Mai 50200, Thailand

**Keywords:** black spot, *Brassica alboglabra*, Sphaerobolaceae, multilocus phylogeny, hormesis, *Bacillus amyloliquefaciens*

## Abstract

Chinese kale (*Brassica alboglabra*) is commonly grown and consumed throughout Asia and is often treated with chemicals to control pests and diseases. In Thailand, public standards, Good Agricultural Practice (GAP), and International Federation of Organic Agriculture Movement (IFOAM) programs were introduced for the cultivation of Chinese kale with minimum input of chemical treatments. Black spot caused by the fungus *Sphaerobolus* has been affecting the plants grown under IFOAM standards in Chiang Mai, Thailand, for several years. Strongly adhering glebal spore masses of the fungus on leaf and stem surfaces have adversely affected postharvest management, especially in the quality classification of the product. Both morphological and phylogenetic (combined ITS, mtSSU, and EF 1-α dataset) studies confirmed a novel species, *S. cuprophilus*. Pathogenicity tests involving inoculation of Chinese kale by non-wound and mulch inoculation bioassays resulted in the production of symptoms of black spot and the re-isolation of *S. cuprophilus*, indicating that the new fungal species is the causal agent of black spots. Inhibitory effects of antagonistic bacteria and chemical fungicides, both allowed for use in plant cultivation under either IFOAM or GAP standards, indicated that *Bacillus amyloliquefaciens* strains (PBT2 and YMB7), chlorothalonil (20 and 500 ppm) and thiophanate-methyl (500 and 1500 ppm) were the most effective in controlling the growth of the causal fungus by 83 to 93%. However, copper oxychloride (5 to 20 ppm), a recommended chemical in control of downy mildew of Chinese kale, showed hormetic effects on *S. cuprophilus* by promoting the growth and sporulation of the fungus. The findings of this study provide vital information regarding the association of *S. cuprophilus* and Chinese kale and will support decisions to manage fungal diseases of this vegetable.

## 1. Introduction

Chinese kale (*Brassica alboglabra* L.H. Bailey), commonly known as Kailaan or Chinese broccoli, is an economic crop widely cultivated and consumed in Asian countries, especially in Thailand [1,2,3]. In addition to good flavor qualities (sweet, crisp, without splinters), Chinese kale contains beneficial nutrients such as glucosinolates, antioxidants, and anticarcinogen compounds [4,5,6,7,8]. Chinese kale is susceptible to several fungal diseases, including black spots caused by *Alternaria brassicicola* and downy mildew caused by *Hyaloperonospora parasitica*, and as such, cultivation in Thailand includes fungicides to control diseases [9,10,11]. However, intensive fungicide application is known to be harmful to human health and other organisms as well as the environment [12,13], and overuse of fungicides results in the development of fungicide resistance in pathogens [14,15]. Chinese kale is grown under the Organic Thailand and International Federation of Organic Agriculture Movement (IFOAM) standard [16]. This is implemented by the Royal Project Foundation, which limits the use of fungicides on agricultural crops. Nonetheless, under the Good Agricultural Practice (GAP) standard, some pesticides can be applied [17].

Chinese kale demonstration plots, as well as farmer fields located in fifteen Royal Project Stations and development centers in Northern Thailand, have been historically used to document diseases. These areas are either under IFOAM or GAP standards, or both, and ongoing collection and identification of disease are performed here regularly. Since 2015, there has been a recovery of an unknown *Sphaerobolus* species that has occurred on the leaves and stems of kale plants growing in one of those fields. Raised black spore masses (glebae) were observed on Chinese kale grown where there had been extensive use of copper fungicides and cow manure. The glebae are discharged by the fungus, which strongly attaches to the surface of the leaves and stems, causing necrosis of the host tissue underneath the glebae (Figure 1A–E). The leaves from affected plants did not reach the minimum requirements of Thai agricultural standards for Chinese kale, especially in which plants must be free of any visible foreign matter or free of pests affecting the general appearance [2,18]. Thus, the black spot caused by the *Sphaerobolus* species is an important postharvest disease that reduces the marketable quality of the stems and leaves of Chinese kale.

The genus *Sphaerobolus* belongs to the family Sphaerobolaceae and the order Geastrales and is normally a saprophytic fungus that has a mechanism for discharging a single spore mass (gleba or peridiole) from its basidiocarp [19,20,21]. *Sphaerobolus* is commonly known as “artillery fungus” because of its capability of ejecting a gleba up to 6 m from its base [19,22,23]. The genus *Sphaerobolus* has been recorded in various continents, including North and South America, Africa, Asia, Australia, and Europe [20,24,25,26,27,28,29,30,31,32]. According to Index Fungorum [33], there are 35 records of the genus *Sphaerobolus*; nevertheless, more recent studies by Geml et al. [20] and Vasava et al. [31] have shown that the genus *Sphaerobolus* consists of four taxonomically distinct species: *S. ingoldii*, *S. iowensis*, *S. stellatus,* and *S. jaysukhianus*. The fungi in the genus *Sphaerobolus* have been considered saprophytes; however, its glebae are responsible for affecting the general appearance of a wide range of plants [34,35] which results in a significant loss in marketable produce. Moreover, *S. stellatus* was confirmed to be a causal agent of thatch collapse [36], whereby the fungus decomposed the lignin in the organic matter below the thatch, thus, resulting in the collapse of the turf.

Unfortunately, some effective fungicides against *Sphaerobolus* species [37] are not permitted under IFOAM or GAP standards. Consequently, the objectives of this study were to: (i) identify and characterize the *Sphaerobolus* species causing black spots, (ii) assess infection and colonization on Chinese kale, and (iii) investigate the antifungal effects of selected antagonistic bacterial strains and fungicides under in vitro conditions.

## 2. Results

### 2.1. Morphological and Physiological Characterization

*Sphaerobolus cuprophilus* sp. nov isolated from the black spot of Chinese kale grew on all tested media at room temperature, albeit at variable growth rates. The highest and most significantly different growth rate, 3.24 to 3.89 mm/day, was found in PDA, half-strength PDA containing 0.2% malt extract, and PDA containing copper oxychloride (5 to 20 ppm) (Table 1). The fungus produced white to creamy white mycelium that developed yellow or orange pigmentation after two to three weeks of incubation. After two weeks, the fungus produced basidiocarps (Figure 2A,B) on OA, PDA, PDA containing cow manure, PDA containing copper oxychloride (5, 10, 20, and 40 ppm), and half-strength PDA containing 0.2% malt extract. Gleba discharge was not found on OA and PDA containing 40 ppm copper oxychloride (Table 1). Basidiocarps opened to reveal dark brown gleba surrounded in gelatinous fluid (Figure 2C,D). Basidiospores are hyaline, elliptical, thick-walled, and have a smooth surface (Figure 2F). Gemmae, cystidia, and basidial chambers were present inside gleba (Figure 2E,G–J). *Sphaerobolus cuprophilus* isolates BS504 and CK1 had similar growth at each temperature, with the highest growth at 30 and 35 °C and the lowest growth at 15 °C (Figure 3).

All species were distinguished based on the presence of basidial chambers, the presence of gammae, gleba size, and other micro-morphological characteristics. The *Sphaerobolus cuprophilus* sp. nov. from Chinese kale differed from other species of *S. ingoldii*, *S. iowensis,* and *S. stellatus* in that it had faster mycelial growth on OA and PDA and larger-sized gleba and basidiospores. *Sphaerobolus cuprophilus* differed from *S. jaysukhianus* in having smaller gleba and the presence of gammae. In addition, the fungus was able to produce cellulase, chitinase, laccase, pectinase, and protease within seven days in appropriate media containing correlated substrates (Table 2). These enzymes might play a significant role in the degradation of plant cell tissue which would enhance colonization by the fungus in Chinese kale.

### 2.2. Phylogenetic Analysis

The combined EF 1-α, mtSSU, and ITS sequences dataset consisted of 3908 total characters (2862 characters were constant, 333 characters were parsimony-uninformative, and 713 characters were parsimony-informative). One hundred equally most parsimonious trees produced by the maximum parsimony (MP) analysis (TL = 1541, CI = 0.842, RI = 0.952, RC = 0.802, and HI = 0.158; see methods) were obtained and compared for the best topology with the Kishino–Hasegawa test. Maximum likelihood (ML; 1000 replicates of bootstrap based on Kimura two-parameter and Nearest-neighbor-interchange heuristic method) and Bayesian probability index (BPI; the Markov chain Monte Carlo method for 10,000,000 generations) were also performed. The phylogenetic tree topologies of MP, ML, and BPI analyses were similar, resulting in the most parsimonious tree (Figure 4). All *Sphaerobolus* species strongly grouped (ML/MP/BPI = 100/100/1) in the same clade and were distinct from the outgroups (*Dacryopinax spathularia* Da3 and *Dacrymyces reseotinctus* Da9pm1). The *Sphaerobolus* clade comprised five species subclades: *S. ingoldii*, *S. iowensis*, *S. stellatus, S. jaysukhianus*, and *S. cuprophilus*. All isolates of *S. cuprophilus* (CK1, BS504, BS504r1, and BS504r2) formed a sister taxon with *S. jaysukhianus* with high support values (ML/MP/BPI = 100/100/1) (Figure 4). The genus *Sphaerobolus* is represented by four species thus far. Here, *S. cuprophilus* sp. nov. is confirmed and described (see Section 2.3.) as a new species based on morphological, cultural characteristics, and the multilocus phylogeny (Figure 2 and Figure 4, Table 2).

### 2.3. Taxonomy

*Sphaerobolus cuprophilus* P. Kalayanamitra & Bussaban, sp. nov. (Figure 2).

MycoBank 847454.

Typification: THAILAND: Chiang Mai, Chom Thong, The Royal Agricultural Station Inthanon, on leaf and stem of *Brassica alboglabra* L.H. Bailey, 3 Dec. 2019, B. Bussaban and S. Nontajak (holotype BBH 49303, dried culture of BS504, deposited at BIOTEC Bangkok Herbarium and Fungarium, National Biobank of Thailand, The National Science and Technology for Development Agency, Pathum Thani, Thailand). GenBank accession numbers: EF 1-α = ON661053; mtSSU = ON133446; ITS = OM980552; LSU = ON053384.

Etymology: cuprophilus means loving copper. This species was able to form abundant basidiocarps in Chinese kale plantation in which copper fungicides has been used.

Description: Colonies on PDA are 19–29 mm in diameter at 7 days at room temperature (24–26 °C). Mycelium is initially cottony white to creamy white, growing slowly with a yellow or orange tint at maturity. Basidiocarps are most often gregarious, occasionally solitary, white, spherical (2.60–) 3.00–5.50 (–6.00) mm in diameter ($\bar{x}$ = 3.90 ± 0.70 mm, *n* = 50). Mature basidiocarps often split stellately, showing yellow to yellow-orange inner layer and gleba. Glebae are brown to dark brown to black, surface rough, with basidiospores, gemmae, and cystidia. Discharged glebae attached on the substrate, slightly deformed (flattened) at the range of 2.00–3.00 mm ($\bar{x}$ = 2.65 ± 0.37 mm, *n* = 20). Basidial chambers are present. Basidiospores are hyaline, elliptical, thick-walled, surface smooth 7.54–10.06 × 5.03–8.80 µm ($\bar{x}$ = 9.05 ± 0.81 × 6.29 ± 1.02 μm, *n* = 50) borne on irregularly scattered basidia in the gleba within the basidial chambers. Gemmae are hyaline, elongated, thin-walled, septate with clamp connection after germination. Cystidia are hyaline, round, surface smooth, 10.06–13.83 µm in diameter ($\bar{x}$ = 12.47 ± 1.00 μm, *n* = 12).

Host: Chinese kale.

Known distribution: Thailand.

Additional collections examined: THAILAND: Chiang Mai, Chom Thong, The Royal Agricultural Station Inthanon, on leaf and stem of *B. alboglabra*, 17 December 2019, S. Nontajak (CK1), GenBank accession numbers: EF 1-α = OP725851; mtSSU = OP595736; ITS = OP584470; LSU = OP594490; Chiang Mai University, on leaf and stem of *B. alboglabra*, 28 January 2020, B. Bussaban (BS504r1), GenBank accession numbers: EF 1-α = ON661054; mtSSU = ON133447; ITS = OM980553; LSU = ON053385; Chiang Mai University, on leaf and stem of *B. alboglabra*, 7 October 2020, P. Kalayanamitra (BS504r2), GenBank accession numbers: EF 1-α = ON661055; mtSSU = ON133448; ITS = OM980554; LSU = ON053386.

Multi-loci (EF 1-α, mtSSU, and ITS) phylogeny: All isolates of *S. cuprophilus* (CK1, BS504, BS504r1, and BS504r2) are monophyletic and formed a sister taxon with *S. jaysukhianus* with high support values (Figure 4).
Key to the species of *Sphaerobolus*1 Basidial chamber absent                        21’ Basidial chamber present                        32 Gammae absent                        *S. ingoldii*2’ Gammae present                       *S. stellatus*3 Gleba size less than 2 mm                     *S. iowensis*3’ Gleba size larger than 2 mm                       44 Gleba size ranges 3.0–3.6 mm, gammae absent          *S. jaysukhianus*4’ Gleba size ranges 2.0–3.0 mm, gammae present         *S. cuprophilus*

### 2.4. Infection and Colonization in Chinese Kale

To fulfill Koch’s postulates, non-wound and mulch inoculation bioassays were performed on six-week-old Chinese kale. One week after non-wound inoculation, water soak spots appeared under the inoculated area and where the epidermis and stomata turned brown (Figure 1F−J). One month after the mulch was inoculated, dark spots appeared on the leaves and stems of colonized Chinese kale. Five months after inoculation, the fungus was still able to discharge glebal masses onto the plants (Figure 1K−M). Symptoms that were detected possessed discharged glebae of *S. cuprophilus* strongly attached to leaves and stems and caused necrosis of the plant tissues (Figure 1N,O). Reisolation from colonized plants was successfully performed and confirmed based on morphological characters and phylogenetic assay. Two retrieved fungal isolates, BS504r1 and BS504r2, were morphologically similar to the original isolates (BS504 and CK1) and fell in the same clade (*S. cuprophilus*) with 77% bootstrap support (MP) and 1.00 Bayesian probability index (Figure 4).

### 2.5. Effects of Selected Bacterial Strains and Fungicides on the Pathogens

Bacteria implemented in biocontrol were tested against *S. cuprophilus* BS504 for inhibitory effects. *Bacillus subtilis* and *B. amyloliquefaciens,* commonly used in Chinese kale farms and included with GAP standard, as well as *Pseudomonas* sp. ZB2 exhibited inhibitory activity against various phytopathogens, including some basidiomycetes [38,39]. All selected *Bacillus* strains obtained from soil (*Bacillus subtilis* CRP3, PBT1, and *B. amyloliquefaciens* PBT2) and *Martianus dermestoides* Fairm 1893 (*B. amyloliquefaciens* YMB7) had the ability to inhibit the fungal growth of *S. cuprophilus* by more than 60%. The percentage of inhibition by both *B. amyloliquefaciens* strains (PBT2 and YMB7) was significantly higher than other strains (80.49 ± 1.89% and 79.27 ± 2.99%, respectively), while that of *Pseudomonas* sp. ZB2 (12.20 ± 7.67%) was significantly lower than other strains (Figure 5A).

Three chemical fungicides allowed to be used in Chinese kale farms under IFOAM and GAP standards (chlorothalonil, copper oxychloride, and thiophanate-methyl) [38] were selected for this study. Thiophanate-methyl at concentrations of 500 ppm and recommended dose on the label (1500 ppm) and chlorothalonil at concentrations of 20 and 500 ppm were the most active against both *S. cuprophilus* isolates (more than 82%) (Figure 5B). Fungal colonies in those plates were effused and grew irregularly with brown sectors. The fungal culture discs in those plates were then transferred onto new PDA plates and incubated at room temperature for determination of the viability of the fungus. All fungicide-treated culture discs were able to grow normally on the PDA plates within 3 days (data not shown). The percentage of inhibition by chlorothalonil ranged from 4.30% to 88.08%. Interestingly, copper oxychloride at concentrations of 5, 10, 20, and 40 ppm and thiophanate-methyl at a concentration of 5 ppm displayed negative inhibition against fungal isolate BS504 (Figure 5B). Moreover, the fungus grown in plates containing 5, 10, and 20 ppm of copper oxychloride ejected more glebal masses than that in the control plates (data not shown).

## 3. Discussion

This is the first global report of the newly described *S. cuprophilus* sp. nov. causing a black spot in Chinese kale. Glebal masses of this species strongly adhered to leaf and stem surfaces affecting the quality of the produce; however, no infection occurred of underlying plant tissue. Hence, affected plants did not reach the minimum requirements of Thai agricultural standards for Chinese kale, especially in which plants must be free of any visible foreign matter or free of pests affecting the general appearance [2,18]. Thus, the black spot caused by *S. cuprophilus* may be considered an important postharvest disease of Chinese kale.

*Sphaerobulus cuprophilus* is distinct from the other existing species based on morphological characteristics and phylogeny. Maximum likelihood, maximum parsimony, and Bayesian analyses of combined datasets (EF 1-α, mtSSU, and ITS) revealed that four isolates of the new species belonged to the genus *Sphaerobolus* and were most closely related to *S. jaysukhianus*. No sequence variation of EF 1-α is detectable within species and low among species within the genus. While low sequence variation of mtSSU and ITS is detectable within species and higher among species within the genus. Especially, ITS regions varied in length from 689 to 1066 bp for *Sphaerobolus* strains investigated, with an insertion of the region in *S. cuprophilus* (847–993 bp) and *S. jaysukhianus* (980–1066 bp) after aligned to other *Sphearobolus* species (689–745 bp). In addition, in the taxonomic key to species of *Sphaerobolus* provided by Vasava et al. [31], the morphology of *S. cuprophilus* matches the description of *S. jaysukhianus*. Both species are similar in having basidial chambers and bigger gleba size than other species. However, the larger basidiospore size and the presence of gemmae of *S. cuprophilus* differentiate this species from *S. jaysukhianus*. A key to species of *Sphaerobolus* provided in this study clarified previously miscategorized *S. ingoldii,* which has basidial chambers [31] and provides a convenient taxonomy for future identification of the *Sphaerobolus* species.

Colonies of newly described *S. cuprophilus* were able to grow and produce basidiocarps on almost all tested culture media at room temperature (24 to 26 °C) and were about one to three times faster than other species grown on OA and PDA. Moreover, *S. cuprophilus* grown in half-strength PDA containing 0.2% malt extract ejected glebae for the first time within 14 days after incubation with natural light. The basidiocarp production of *S. ingoldii*, *S. iowensis,* and *S. stellatus* were found abundantly on wheat straw agar [20], whereas no basidiocarps of *S. cuprophilus* developed on rice straw agar. PDA containing cow manure or copper oxychloride promoted the growth and sporulation of *S. cuprophilus*, which was consistent with the environment in local Thai fields. Rice straw itself might not directly promote fungal growth or basidiocarp formation, but the mulch provides optimum conditions which may support mycelial growth and spore dispersal [22,40].

*Sphaerobolus cuprophilus* did not rapidly infect and colonize the plant epidermis of leaf and stem tissues, nor did it produce enlarged lesions. However, the tissues, as well as stomata underneath the colonized area of Chinese kale, became necrotic in both naturally and artificially inoculated plants. This may be due to the glebae of *S. cuprophilus* consisting of thin-walled gemmae, which can easily germinate. After the germination of the gemmae, colonization of the substrate begins. *Sphaerobolus cuprophilus* was able to produce various enzymes, including cellulase and pectinase. Hence it is likely that the colonization by *S. cuprophilus* is due to the production of these enzymes since cellulose, hemicellulose, and pectin are most of the major primary cell wall components of dicotyledonous plants, including kale. Exoenzyme production by *S. cuprophilus* was also consistent with previous studies of the white-rot fungus *S. stellatus* [41,42] that can be attributed to the digestion of cellulose, hemicellulose, and lignin in the substrate by the production of enzymes such as lignolytic enzymes, cellulase, peroxidase, laccase, and xylanase.

All *Bacillus* strains isolated from soil and a medicinal insect, *Martianus dermestoides* Fairm 1893, showed antagonistic activity against *S. cuprophilus. Pseudomonas* sp. ZB2, however, showed a negative result. *Bacillus subtilis* and *B. amyloliquefaciens* are commonly used for the control of various pathogens such as *Alternaria brassicicola*, *Hyaloperonospora parasitica*, *Xanthomonas campestris* pv. *campestris*, etc., on Chinese kale [38]. According to Kodchasee [39], these *Bacillus* strains could grow in a wide range of temperatures, 25 to 50 °C, and on nutrient agar containing benomyl and mancozeb. Besides, they are known to produce high amounts of auxins and siderophores, which are secondary metabolites that are important for promoting plant growth. Auxins are one of the plant-growth regulators that have an important role in the plant immune system [43,44]. At the same time, siderophores are iron-chelating compounds that are involved in nutrition competition and plant defense [45]. Even though *Pseudomonas* spp. secrete siderophores [46], *Pseudomonas* sp. ZB2 grew much slower than *Bacillus* strains, especially on PDA. This could imply that PDA might have an impact on the percentage of growth inhibition in *Pseudomonas* sp. ZB2. Thiophanate-methyl (500 and 1500 ppm recommended dose from the label) and chlorothalonil (20 and 500 ppm) were the most effective in suppressing the growth of *S. cuprophilus* under in vitro conditions; however, thiophanate-methyl at low concentration (5 ppm) stimulated fungal growth. In contrast, decreased inhibitory effects were found on media that contained 40 ppm of chlorothalonil. This observation indicated that chlorothalonil and thiophanate-methyl might have in vitro hormetic effects on *S. cuprophilus*. Hormesis to fungicides (e.g., carbendazim, thiophanate-methyl) in some pathogenic fungi, such as *Botrytis cinerea* and *Sclerotinia homoeocarpa*, have recently been reported [47,48]. The effects may increase pathogen virulence under periods of sub-lethal fungicide exposure and result in increased disease incidence and severity [49]. Therefore, fungicide-induced hormesis in the phytopathogen should be determined in in vivo assays. Future field research using thiophanate-methyl should also be conducted since forms of this fungicide are registered for wide use in Thailand and allowed for use in plant cultivation under the GAP standard [38].

Most concentrations of copper oxychloride (5 to 20 ppm) not only showed negative results in the control of vegetative growth of *S. cuprophilus*, but they also promoted basidiocarp production and maturation. Galhaup et al. [50], Liu et al. [51], Tychanowicz et al. [52], and Zhuo et al. [53] demonstrated that copper increased the laccase production of many white-rot fungi. Generally, laccase is found in plants, bacteria, insects, and fungi and plays different roles in different organisms [54]. In fungi, laccase is associated with morphogenesis [55] and plant pathogenesis [54,56]. Laccase genes have also been shown to have complex functions in *Setosphaeria turcica*, an important causal agent of northern corn leaf blight [57,58,59]. There are nine laccase genes in the genome of *S. turcica* reported with three main expressed active laccases, *StLAC1*, *StLAC2*, and *StLAC6* [60]. The latter two laccase genes play a role in the pathogenicity of *S. turcica* but in a completely different way. *StLAC2* is involved in melanin content reduction of the gene knockout mutants, and the conidia of mutants cannot be produced; these directly affect the infection ability of the fungus [57]. While *StLAC6* has been shown to be involved in peroxisome function and pathogenicity by participating in the synthesis of phenolic metabolites but has no effect on the growth or development of the fungus. The deletion of *StLAC6* also led to an increase in the fungicide sensitivity (e.g., boscalid, pyraclostrobin, and tricyclazole) and the differential expression of other laccase genes [59]. In this study, *S. cuprophilus* was found to have the ability to produce laccase enzyme (Table 2); thus, it is possible that copper metal in the chemical had the potential to induce laccase and basidiocarp production of the fungus. Nevertheless, there has been only one report regarding whole genome sequencing of *Sphaerobolus.* Hence, more research on the mitogenome or whole genome sequencing, especially laccase involving genes, of *Sphaerobolus* species is required to further the understanding of the laccase enzyme and its role in the fungus. Copper ion is required as a micronutrient for fungal growth and proliferation, but in excess conditions, it can be potentially toxic [61]. In addition, fungicides such as copper oxychloride and copper hydroxide have been applied to successfully control downy mildew in Chinese kale grown under the IFOAM standard by the Royal Project Foundation farmers. *Sphaerobolus cuprophilus* abundantly formed basidiocarps in Chinese kale grown in the same cultivation area. Based on yearly field observations, hormesis to this fungicide in *S. cuprophilus* has already occurred in the field.

*Sphaerobolus cuprophilus* sp. nov. is a potential postharvest pathogen that can affect the quality of Chinese kale. Black spot (glebae attachment) and water-soaked spot symptoms on Chinese kale grown in fields and in pathogenicity tests showed that the fungus was not a necrotrophic pathogen. Preliminary results suggested that *B. amyloliquefaciens* strains PBT2 and YMB7, thiophanate-methyl, and chlorothalonil, which are allowed for use in fields under either IFOAM or GAP standards, were the most effective. In addition, all selected *Bacillus* strains could be used as biological control agents (% inhibition > 60) and/or could be used together with some fungicides to optimize the inhibition as well as reduce the use of chemical fungicides. Consequently, further research is required to study the effects of these antagonistic bacterial strains and fungicides in vivo to determine whether the results presented here will be consistent in field trials.

## 4. Materials and Methods

### 4.1. Sample Collection and Fungal Isolation

Leaves and stems with dark spot symptoms were collected from Chinese kale grown under the IFOAM standard by the Royal Project Foundation farmers in Chom Thong, Chiang Mai, Thailand (18.32 N, 98.31 E) in December 2019. Four symptomatic tissues from four plants were cleansed by cotton plugs soaking with 70% ethanol, air dried, and pieces of glebae were aseptically placed onto potato dextrose agar (PDA). The cultures were incubated at room temperature (24 to 26 °C) until mycelium and basidiocarps developed. Pure fungal cultures were stored in PDA slant agar and liquid paraffin at room temperature for further studies. The fungal herbarium was deposited in the BIOTEC Bangkok Herbarium and Fungarium (BBH), National Biobank of Thailand, The National Science and Technology for Development Agency, Pathum Thani, Thailand.

### 4.2. Morphological and Cultural Characterization

Fungal isolates were selected for morphological and cultural characterization. Fungal isolates were grown at room temperature (24 to 26 °C) on oatmeal agar (OA), PDA, half-strength PDA containing 0.2% malt extract, PDA containing 0.05% amoxycillin, PDA containing 4 g of cow manure (Agriculture CMU, Chiang Mai, Thailand), PDA containing copper oxychloride (5, 10, 20, 40, 500, and 1000 ppm) and rice straw agar (RSA; adapted from Geml et al. [20]). The culture discs (6 mm in diameter) were prepared by cutting the edge of 14-day-old fungal colonies and placing one disc at the center of each Petri dish, with six replicates for each medium. Inoculated plates were then incubated at room temperature (24 to 26 °C) with either natural light or 12 h-fluorescence light (3047 lux) for two weeks. To determine the optimum temperature for the fungus, the growth rates of two selected isolates (BS504 and CK1) were studied on PDA at 15, 20, 25, 30, and 37 °C. The culture disc (6 mm in diameter) was placed at the center of each PDA plate. Four replicates of each isolate were sealed with parafilm and incubated at temperatures mentioned earlier. Growth of each culture was observed daily, and growth rates were calculated after three weeks of incubation. Morphological characteristics such as discharged glebae, basidiospores, and cystidia were measured under either the Olympus SZ40 stereo microscope (Olympus Corporation, Tokyo, Japan) or Olympus CX31 compound microscope (Olympus Corporation, Tokyo, Japan).

The ability of the fungus to produce enzymes (cellulase, chitinase, laccase, lipase, pectinase, and protease) was studied on six different media containing correlated substrates of carboxymethyl cellulase agar [62], Chitinase detection agar [63], N-limited Kirk’s agar [64], tributyrin agar [65], modified pectin agar [62], and skim milk agar (HiMedia Laboratories, Nashik, India). Fungal culture discs (6 mm in diameter) were similarly prepared as described above and placed onto the surface of each agar plate. Four inoculated plates of each medium were then incubated at appropriate conditions for 7 to 14 days, and enzyme activity was determined according to previous reports [62,63,64,65].

The growth rate of each isolate was compared across temperatures by one-way analysis of variance (ANOVA), and the growth rate was compared among isolates at each temperature by independent-sample T-test using SPSS statistics v.25 (IBM, Armonk, NY, USA). Data on other cultural characteristics and enzyme activities were analyzed by one-way ANOVA. Duncan’s multiple range test (*p *< 0.05) was used to detect significant differences in means.

### 4.3. DNA Extraction, PCR Amplification, and Sequencing

Pure fungal mycelium of four isolates was stored at −20 °C in 100 µL of buffer A (TOYOBO, Osaka, Japan). For DNA extraction, the fungal mycelium in buffer A was ground with quartz sand and centrifuged at 8000 rpm at 4 °C for 10 min. Fifty microliters of template DNA was transferred into a new microcentrifuge tube. Mitochondrial small ribosomal subunit gene (mtSSU), nuclear large ribosomal subunit gene (LSU), translation elongation factor 1-α gene (EF 1-α), and the entire internal transcribed spacer (ITS) gene were amplified by different primer pairs: ITS5 and ITS4 for ITS; NL1 and NL4 for LSU; MS1 and MS2 for mtSSU; EF1-983F and EF1-1567R for EF 1-α [20,66,67]. All PCR reaction mixtures consisted of 12.5 µL of PCR buffer for KOD FX Neo (TOYOBO, Osaka, Japan), 5.0 µL of dNTPs, 0.75 µL of forward and reverse primers, 0.5 µL of KOD FX Neo, 4.75 µL of PCR grade water and 1.0 µL of template DNA. PCR amplification was performed as described by Geml et al. [20]. PCR products were stained with MaestroSafe Nucleic Acid Stains (Maestrogen, Hsinchu City, Taiwan) and visualized under LED light on 1.5% agarose gel. PCR products were purified using TIANquick Midi Purification Kit (TIANGEN BIOTECH, Beijing, China), following the manufacturer’s protocol, and then sequenced by Macrogen (Seoul, South Korea), using primers described above except for the EF 1-α gene, which was replaced by EF1-1567Ra primer for better results [20].

### 4.4. Phylogenetic Analysis

Forward and reverse sequences of four isolates were assembled in Sequencher 4.8 for Windows (Gene Codes, Ann Arbor, MI, USA) and used to query via BLAST [68]. Sequences of *Sphaerobolus* from Chinese kale and the outgroup taxa which belong to the same subphylum as *Sphaerobolus* but different class (*Dacrymyces reseotinctus* Da9pm1 and *Dacryopinax spathularia* Da3) were deposited in GenBank. Reference *Sphaerobolus* type and epitype species and voucher isolates were also obtained from GenBank (Table 3). Multiple sequence alignments of each locus were generated using MAFFT online version [69] and deposited in TreeBASE (S29856). Phylogenetic trees were constructed by PAUP* 4.0b10 [70]. The combined dataset was performed based on maximum parsimony (MP), maximum likelihood (ML), and Bayesian analyses (Markov Chain Monte Carlo; MCMC). For MP analysis, heuristic searches with a tree-bisection-reconnection branch-swapping algorithm were performed. Starting trees were obtained via stepwise addition with 100 random sequence input orders. The parsimony tree scores, including tree length (TL), consistency index (CI), retention index (RI), rescaled consistency (RC), and homoplasy indices (HI), were calculated. To test the statistical reliability of the generated trees and test the stability of clades, the bootstrap test [71] was used with 1000 replicates. For ML and Bayesian analyses, appropriate approaches available in MEGA X [72] and BEAST v.1.10.4 [73] were performed, respectively. The bootstrap supports of MP and ML (≥50%) and Bayesian probability index (≥0.99) were also given above the tree branches.

### 4.5. Pathogenicity Test

Chinese kale seeds of cultivar Ocean KA 031 (Chia Tai, Bangkok, Thailand) were surface sterilized with 1% (ai) sodium hypochlorite for 5 min and rinsed with sterile water three times. The seeds were then soaked in warm water (60 °C) for 20 min and planted in autoclaved potting soil mixed with rice charcoal (1:1, pH ≈ 6.0) for 20 days before being transferred into new plastic pots (14.5 cm in diameter of the top; 12 cm tall) containing autoclaved potting soil mixed with rice charcoal (2:1, pH ≈ 6.0). The mulch of rice straw was placed on top of the pots to reduce moisture loss and keep weeds controlled. The plants were watered twice daily and fertilized with cow manure and 15-15-15 NPK fertilizer (Viking Fertilizer, Nonthaburi, Thailand) 15 days after transplanting. Fungal culture discs (5 mm in diameter of 14-day old on PDA at 24 to 26 °C) or PDA discs (control) were inoculated onto the second and the third leaves (1 disc/leaf) from shoots of each plant after surface sterilization of the leaves with 70% ethanol. All six treated plants were incubated in the plastic box (45 cm × 30 cm × 45 cm) covered with a porous lid to maintain high relative humidity and observed daily for symptom appearance. In addition, the mulch was also inoculated with mycelial discs, and the mulch inoculated with PDA discs was used as a control. The inoculation was done by placing ten mycelial discs underneath the mulch. All twelve treated plants were incubated in the greenhouse with a 12-h photoperiod and observed daily for symptom appearance. Ejected glebae attached to Chinese kale leaves and stems were studied under compound and stereo microscopes. Morphological and cultural characteristics of reisolated cultures (BS504r1 and BS504r2) were identified and compared with the original fungus.

### 4.6. Effects of Selected Bacterial Strains and Fungicides on Sphaerobolus cuprophilus sp. nov.

Five bacterial strains that exhibited high antagonistic activity against various phytopathogens according to Kodchasee [39] (*Bacillus subtilis* CRP3, *B. subtilis* PBT1, *B. amyloliquefaciens* PBT2, *B. amyloliquefaciens* YMB7, and *Pseudomonas* sp. ZB2) were tested against *S. cuprophilus* BS504 isolated from Chinese kale, using dual culture method. Fungal mycelial discs from actively growing cultures were placed at the center of fresh PDA plates. All bacterial inoculants were prepared by culturing individually in nutrient broth (NB) at 90 rpm for 24 h in a rotary shaker (Tokyo Rikakikai, Tokyo, Japan). Each bacterial suspension was adjusted to 0.5 McFarland standard turbidity and then inoculated 2.5 cm alongside the fungal culture discs. Control treatment contained only the culture discs at the center. Each treatment was replicated six times and incubated at room temperature (24 to 26 °C) for 14 days. The radius of each colony was measured (mm), and the percentage of inhibition was calculated as follows:(1)$$\%\;\mathrm{inhibition}=\left[\frac{\left(R1\;-\;R2\right)}{R1}\right]\times 100$$
where *R*1 is the average radius of the fungal colony in dual culture and *R*2 is the average radius of the fungal colony in control plates.

Studying the efficacy of fungicides on growth inhibition of fungal isolates BS504 and CK1 was conducted with chlorothalonil (Extra Agrochemical, Samut Prakan, Thailand), copper oxychloride (Erawan Agricultural chemicals, Bangkok, Thailand) and thiophanate-methyl (T.J.C. Chemical, Bangkok, Thailand), using poisoned food technique [74]. The fungal culture discs were placed at the center of PDA plates containing twofold serial dilutions of each fungicide ranging from 5 to 40 ppm, 500 ppm, together with the recommended dose as described on the fungicide labels. Control treatment contained no fungicides. Each treatment was replicated at least six times and incubated at room temperature (24 to 26 °C) for three weeks. The diameter of each colony was measured (mm), and the percentage of inhibition was calculated as follows:(2)$$\%\;\mathrm{inhibition}=\left[\frac{\left(D1\;-\;D2\right)}{D1}\right]\times 100$$
where *D*1 is the diameter of the fungal colony in plates with fungicide and *D*2 is the diameter of the fungal colony in control plates.

The experiment was repeated twice. A significant difference between the percentage of growth inhibition was evaluated with Duncan’s multiple range test (*p *< 0.05) using SPSS statistics.

## Figures and Tables

**Figure 1 plants-12-00480-f001:**
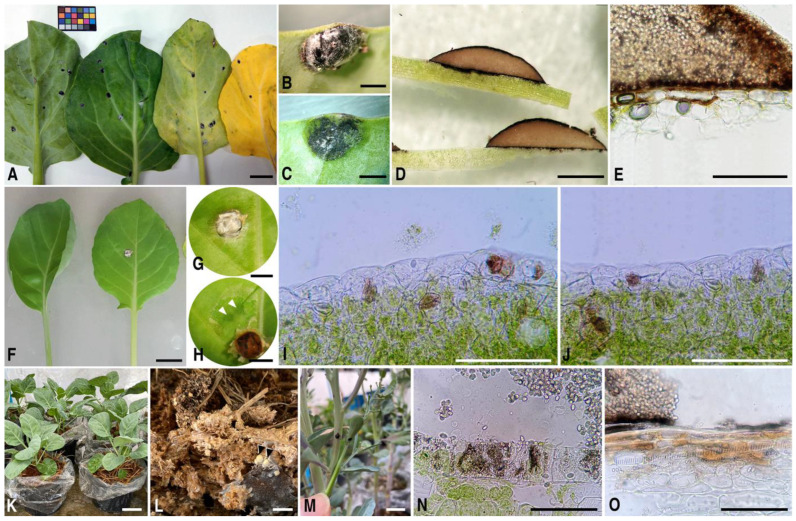
Black spot of Chinese kale caused by *Sphaerobolus cuprophilus*. (**A**–**E**) Symptoms on naturally infected plants; (**A**) Discharged glebae attached on leaves producing black spots; (**B**) Top view of black spot on the leaf; (**C**) Leaf tissue underneath the black spot in (**B**); (**D**,**E**) Sections of gleba and necrotic tissue; (**F**–**O**) Pathogenicity tests of *S. cuprophilus* on Chinese kale; (**F**–**H**) Symptoms on leaves after non-wound inoculation; (**I**,**J**) Cross-section of inoculated leaves; (**K**) Kale plants growing in inoculated mulch; (**L**) Mycelium and basidiocarps (arrowed) of *S. cuprophilus* developed on mulch; (**M**) Discharged glebae on the plant leaves and stems; (**N**) Cross-section of necrotic leaves; (**O**) Cross-section of necrotic stems. Scale bars: (**A**,**F**,**L**,**M**) = 10 mm; (**B**–**D**) = 1 mm; (**E**,**I**,**J**,**N**,**O**) = 100 µm; (**G**,**H**) = 5 mm; (**K**) = 5 cm.

**Figure 2 plants-12-00480-f002:**
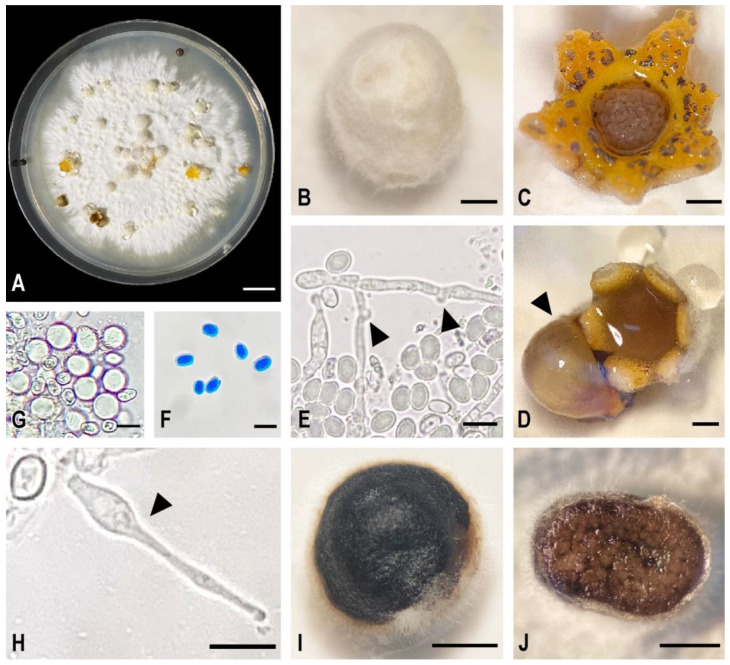
Morphological characters of *Sphaerobolus cuprophilus*. (**A**) Colony on potato dextrose agar incubated at 25 °C for 4 weeks; (**B**) Unopened basidiocarp; (**C**) Opened basidiocarp with gleba; (**D**) Opened basidiocarp with an everted elastic membrane (arrowed) after discharging the gleba; (**E**) Basidiospores and germinated gemmae with clamp connections (arrowed); (**F**) Basidiospores stained in cotton blue; (**G**) Elliptical basidiospores and spherical cystidia; (**H**) Thick-walled basidiospore and thin-walled, elongated gemma (arrowed); (**I**) Discharged gleba; (**J**) Basidial chambers. Scale bars: (**A**) = 1 cm; (**B**–**D**,**I**,**J**) = 1 mm; (**E**–**H**) = 10 µm.

**Figure 3 plants-12-00480-f003:**
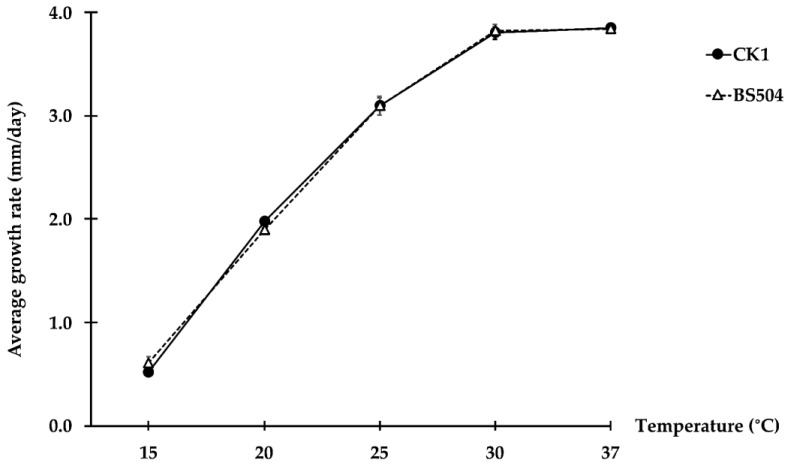
Average growth rates of *Sphaerobolus cuprophilus* isolates CK1 and BS504 on potato dextrose agar at different temperatures.

**Figure 4 plants-12-00480-f004:**
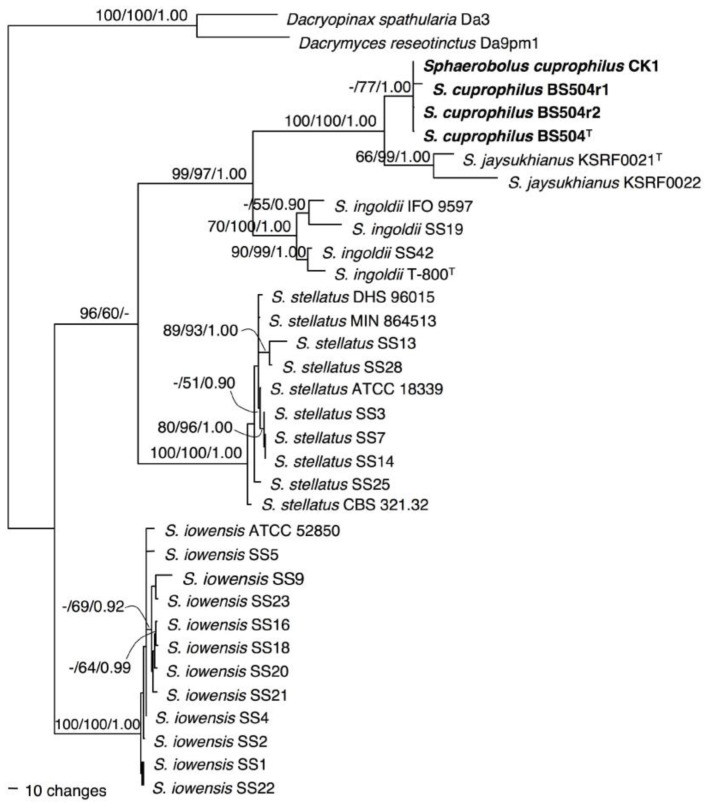
Maximum parsimony phylogenetic tree of the combined ITS, mtSSU, and EF1-α dataset of *Sphaerobolus cuprophilus* and related species. *Dacryopinax spathularia* Da3 and *Dacrymyces reseotinctus* Da9pm1 were used as the outgroup taxa. The bootstrap values of maximum likelihood and maximum parsimony (>50%) representing 1000 bootstrap replications and Bayesian probability index are indicated above the branches. Type or ex-type strains of each taxon are superscript with T.

**Figure 5 plants-12-00480-f005:**
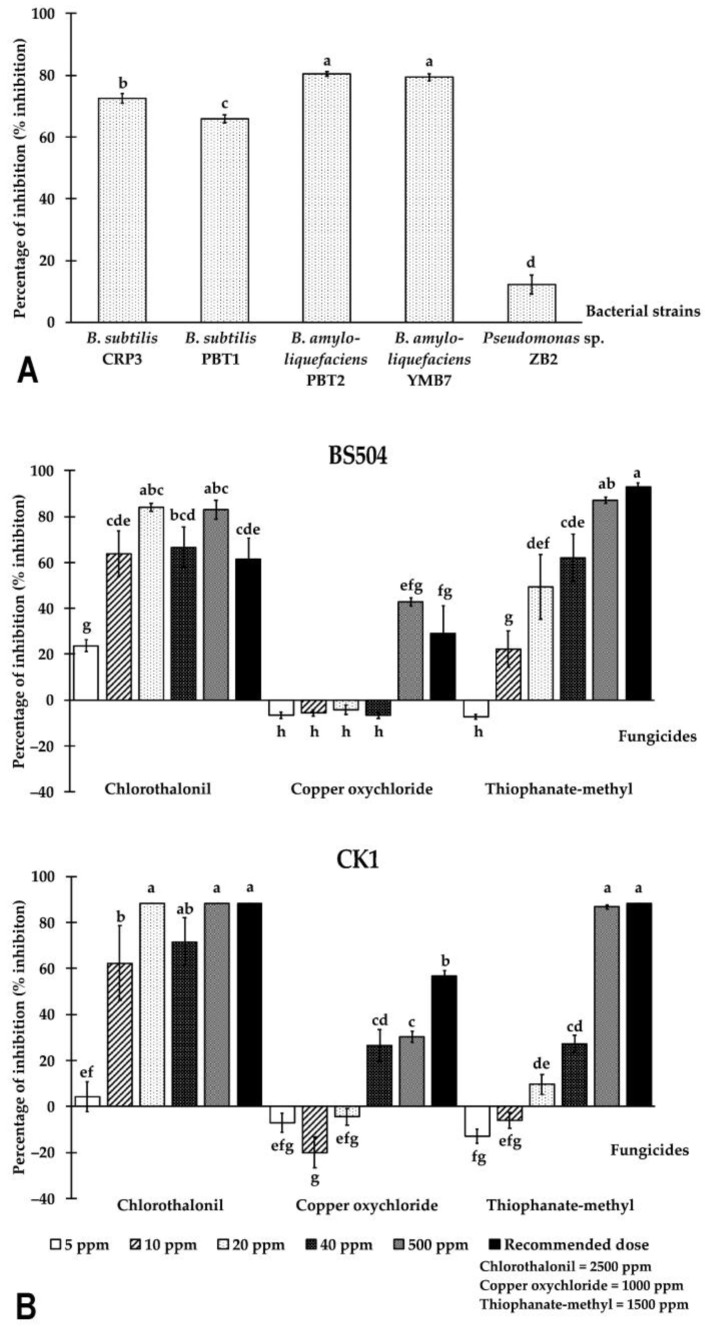
(**A**) Percentage of inhibition by selected antagonistic bacteria against *Sphaerobolus cuprophilus* BS504; (**B**) Percentage of inhibition by chemical fungicides at different concentrations on the mycelial growth of *S. cuprophilus* isolates BS504 and CK1. The data presented are the mean of at least six replicates ± standard error of means. Different letters indicate treatments significantly different (*p *< 0.05) according to Duncan’s multiple range test.

**Table 1 plants-12-00480-t001:** The growth rate of *S. cuprophilus* BS504 on different culture media incubated at room temperature (24–26 °C).

Culture Medium	Growth Rate (mm/Day) *	Basidiocarp Formation	Gleba Discharge (1st Discharge, Dai)	Gleba Size (mm)
Oatmeal agar (OA)	3.07 b	+	−	−
Potato dextrose agar (PDA)	3.24 ab	+	+(21)	2.65 abc
Half-strength PDA + 0.2% ME	3.89 a	+	+(14)	2.58 bc
PDA + 0.05% amoxicillin	3.10 b	−	−	−
PDA + cow manure	3.48 ab	+	+(22)	3.15 a
PDA + 5 ppm copper oxychloride	3.40 ab	+	+(21)	2.33 c
PDA + 10 ppm copper oxychloride	3.51 ab	+	+(21)	2.50 bc
PDA + 20 ppm copper oxychloride	3.32 ab	+	+(21)	3.07 ab
PDA + 40 ppm copper oxychloride	3.07 b	+	−	−
PDA + 500 ppm copper oxychloride	1.96 c			
PDA + 1000 ppm copper oxychloride	1.96 c	−	−	−
Rice straw agar	1.60 c	−	−	−

* Each value is the mean of at least six replicates. Values within the same column followed by different lowercase letters are significantly different at *p *< 0.05, according to Duncan’s multiple range test. Dai: Day after incubation.

**Table 2 plants-12-00480-t002:** Comparison of morphological characteristics and enzyme production of *Sphaerobolus* species.

Characteristics	*Sphaerobolus cuprophilus* P. Kalayanamitra & Bussaban sp. nov.	*S. ingoldii* J. Geml, D.D. Davis & D.M. Geiser	*S. iowensis* L.B. Walker	*S. jaysukhianus* A.M. Vasava, R.S. Patel & K.S. Rajput	*S. stellatus* (Tode) Pers.
Basidial chambers	Present	Absent	Present	Present	Absent
Gemmae	Present	Absent	Present	Absent	Present
Gleba size (mm)					
Mean	2.65 ± 0.37	0.98 ± 0.20	1.55 ± 0.16	3.5	1.50 ± 0.29
Range	2.00–3.00	0.60–1.30	1.20–1.60	3.00–3.60	Nd
Basidiospore size (µm)					
Mean	9.05 ± 0.81 × 6.29 ± 1.02	8.78 ± 0.76 × 5.87 ± 0.58	7.22 ± 0.51 × 4.92 ± 0.22	Nd	7.27 ± 0.50 × 4.57 ± 0.29
Range	7.54–10.06 × 5.03–8.80	8.00–9.50 × 5.40–6.20	6.00–10.00 × 5.00–6.00	8.50–10.80 × 5.30–7.30	Nd
Growth rate (mm/day)					
Potato dextrose agar (PDA)	3.24 ± 0.45	1.35 ± 0.43	1.03 ± 0.13	Nd	1.10 ± 0.48
Oatmeal agar (OA)	3.07 ± 0.04	2.10 ± 0.04	1.31 ± 0.24	Nd	1.38 ± 0.36
Enzyme production	Cellulase, chitinase, laccase, lipase, pectinase, protease	Nd	Nd	Nd	Cellulase, laccase, Mn-independent peroxidase, mannanase, xylanase

Nd: No data available.

**Table 3 plants-12-00480-t003:** Fungal isolates and GenBank accession numbers used in the phylogenetic analysis.

Species	Isolate	Geographic Origin	GenBank Accession Number			
			EF 1-α	mtSSU	ITS	LSU
*Dacrymyces reseotinctus*	Da9pm1	Chiang Mai, Thailand	ON661052	ON133445	OM980556	ON053383
* Dacryopinax spathularia *	Da3	Chiang Mai, Thailand	ON661051	ON133444	OM980555	ON053382
* Sphaerobolus cuprophilus *	BS504 ^T^	Chiang Mai, Thailand	ON661053	ON133446	OM980552	ON053384
	BS504r1	Chiang Mai, Thailand	ON661054	ON133447	OM980553	ON053385
	BS504r2	Chiang Mai, Thailand	ON661055	ON133448	OM980554	ON053386
	CK1	Chiang Mai, Thailand	OP725851	OP595736	OP584470	OP594490
* S. ingoldii *	T-800	Kellogg Biological Station Long Term Ecological Research, Michigan	AY654734	AY654739	AY654737	AF139975
	IFO 9597	Inst. For Fermentation, Otsu, Japan	AY487996	AY488022	AY487971	AY439013
	SS19	Atlanta, Georgia	AY487990	AY488015	AY487965	AY439012
	SS42	Hershey, Pennsylvania	AY654735	AY654740	AY654738	
* S. iowensis *	ATCC 52850	East Lansing, Michigan	AY487984	AY488008	AY487958	AY439014
	SS1	Indiana	AY487976	AY488000	AY487950	
	SS2	Elizabethtown, Pennsylvania	AY487977	AY488001	AY487951	
	SS4	Langhorne, Pennsylvania	AY487979	AY488003	AY487953	
	SS5	State College, Pennsylvania	AY487980	AY488004	AY487954	
	SS9	Chapel Hill, North Carolina	AY487982	AY488006	AY487956	AY439010
	SS16	Olney, Maryland	AY487988	AY488012	AY487962	
	SS18	Olney, Maryland	AY487989	AY488014	AY487964	
	SS20	Olney, Maryland	AY487991	AY488016	AY487966	
	SS21	Galion, Ohio	AY487992	AY488017	AY487967	
	SS22	Ithaca, New York	AY487993	AY488018	AY487968	
	SS23	Medina, Ohio	AY487994	AY488019	AY487969	
* S. jaysukhianus *	KSRF0021 ^T^	Vadodara, Gujarat, India	MK231137	MK208481	MK208479	MK208480
	KSRF0022	Gujarat, India	MK243684	MK209117	MK209116	MK209118
* S. stellatus *	ATCC 18339	Maryland	AY487983	AY488007	AY487957	AY439011
	CBS 321.32	The Netherlands	AY487999	AY488026	AY487975	
	DSH 96-015	Great Brook State Park, Massachusetts	AY487985	AY488009	AY487959	
	MIN 864513	Elm Creek Nature Reserve, Minnesota	AY487998	AY488025	AY487974	
	SS3	State College, Pennsylvania	AY487978	AY488002	AY487952	
	SS7	West Mifflin, Pennsylvania	AY487981	AY488005	AY487955	
	SS13	Erie, Pennsylvania	AY487986	AY488010	AY487960	
	SS14	Lucinda, Pennsylvania	AY487987	AY488011	AY487961	
	SS25	Newton Centre, Massachusetts	AY487995	AY488021	AY487970	
	SS28	Anchorage, Alaska	AY487997	AY488024	AY487973	

^T^ = type or ex-type strain.

## Data Availability

Not applicable.

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
