# Peer review of "Identification, Characterization, and Control of Black Spot on Chinese Kale Caused by Sphaerobolus cuprophilus sp. nov."

_plants, 2023, doi:10.3390/plants12030480_

Round 1

Reviewer 1 Report

The manuscript "Identification, Characterization, and Control of Black Spot on Chinese Kale Caused by Sphaerobolus cuprophilus sp. nov." was excellent, but I believe it would be better suited to the JOF journal.Over all, the manuscript had some minor edits.

1. The introduction needs to be more accurate and full of potential. Add more work.

2: All the scientific names need to be revised; some of them are not italic. plz revise

3: All the images were of very poor resolution. Please provide us with high-quality images. 

Author Response

  1. The authors have uploaded higher resolution of all Figures in a zip file and really apologise for a lower resolution of those Figures in the previous manuscript since we was following the guide for authors--limitation of file size of manuscript for uploading in the submission system.
  2. The authors have corrected English grammar follow all reviewers.
  3. The authors have followed all reviewers to revise all major editorial comment on the attached pdf for author's consideration (e.g., italic all scientific names, adding more work in ‘Introduction’)

Author Response

  1. The authors have uploaded higher resolution of all Figures in a zip file and really apologise for a lower resolution of those Figures in the previous manuscript since we was following the guide for authors--limitation of file size of manuscript for uploading in the submission system.
  2. The authors have corrected English grammar follow all reviewers.
  3. The authors have followed all reviewers to revise all major editorial comment on the attached pdf for author's consideration (e.g., putting error bars in Figure 3, discussing how similar of the combined sequences of the S. cuprophilus isolates and other species, adding citation for GAP and IFOAM, expanding more details around Koch’s postulates in the ‘Materials and Methods’, and emphasising in the ‘Introduction’ that Sphaerobolus is regarded as a post-harvest pathogen and in the ‘Discussion’ that pathogenicity tests showed that S. cuprophilus was not a necrotrophic pathogen).

Round 2

Reviewer 1 Report

The authors addressed my comments very well. I think the manuscript could be accepted now. 

Author Response

The authors have double checked for English language and grammatical errors.

Author Response

  1. The authors have double checked for English language and grammatical errors.
  2. The authors have followed all reviewers to revise all minor editorial comment on the attached pdf for author's consideration (e.g., breaking of some sentences for easier following, adding more work from some literature regarding the role of laccase in ‘Discussion’).
  3. The authors would like to keep the colour swatch in the image in panel A of Figure 1 since it does not hinder and it can demonstrate the colour of the symptom, black spot.